# Nitrogen and Phosphorus Co-Fertilization Affects *Pinus yunnanensis* Seedling Distribution of Non-Structural Carbohydrates in Different Organs After Coppicing

**DOI:** 10.3390/plants14030462

**Published:** 2025-02-05

**Authors:** He Sun, Yu Wang, Lin Chen, Nianhui Cai, Yulan Xu

**Affiliations:** 1Key Laboratory of National Forestry and Grassland Administration on Biodiversity Conservation in Southwest China, Southwest Forestry University, Kunming 650224, China; hesun0202@gmail.com (H.S.); wangyu@163.com (Y.W.); linchen@swfu.edu.cn (L.C.); cainianhui@swfu.edu.cn (N.C.); 2Key Laboratory of Forest Resources Conservation and Utilization in the Southwest Mountains of China, Ministry of Education, Kunming 650224, China

**Keywords:** *Pinus yunnanensis*, coppicing, non-structural carbohydrate, fertilization, nitrogen, phosphorus

## Abstract

The effects of nutritional additions on the non-structural carbohydrates (NSCs) of *Pinus yunnanensis* Franch. following coppicing were examined in this work. Three levels of phosphorous (P) addition, namely P_0_ (0 g/plant), P (2 g/plant), and P+ (4 g/plant), and two levels of nitrogen (N) additions, namely N0 (0 g/plant) and N+ (0.6 g/plant) The treatments consisted of D1 (N_0_P), D2 (N+P_0_), D3 (N_0_P), D4 (N+P), D5 (N_0_P), and D6 (N+P+), utilizing an orthogonal design to assess how these nutrients influence NSC levels and their components throughout many plant organs in *P. yunnanensis*. The findings showed that fertilization enhanced NSCs and their components’ contents in *P. yunnanensis*. P treatment greatly raised NSC levels in sprouts as well as starch (ST) content in stems and sprouts. N treatment greatly raised soluble sugar (SS) and NSC content in stems and greatly accelerated the contents of NSCs and their components in sprouts. The combined application of N and P further improved SS content in stems. Fertilization effects varied over time, with significant increases in NSC content observed at different stages: at 0 d, fertilization significantly raised NSCs and their components in needles; at 90 d, roots and stems showed increases in both NSCs and their components’ contents; at 180 d, stem ST content significantly increased; and at 270 d, NSCs and their components’ contents across all organs were significantly increased. Especially in roots, stems, and sprouts, the combined N (0.6 g/plant) and P (2.0 g/plant) treatment (D4) produced the highest NSC concentration among the treatments. This suggested that NSC formation in plants might be greatly promoted by a balanced N and P fertilization ratio acting in concert. Moreover, fertilizer, as part of a general management plan, has long-term and significant benefits on plant development, especially after coppicing, accelerating recovery, expanding growth potential, and thereby strengthening the plant’s capacity to adapt to environmental changes.

## 1. Introduction

Nitrogen (N) and phosphorous (P) are among the rare elements that have a significant effect on the carbon cycle [1]. While targeted soil fertilization catered to particular nutrient needs can effectively promote strong seedling development, the application of N and P fertilizers was vital for stimulating seedling development in daily life. N fertilizer, especially, not only promotes seedling development but also affects the P concentration in the soil. P fertilizers affect seedling growth by changing the soil microenvironment and promoting root development [2,3]. Previous studies have shown that plants suffer when deprived of vital nutrients such N and P [4,5,6], and that photosynthesis regularly shows a decrease when N and P deficits exist [7]. The increase in plant growth limitation is attributed to deficiencies in N and P [8]. The rational application of fertilizer is essential for meeting the demand for high and stable yields, while simultaneously preventing wastage of fertilizer and environmental pollution [9].

With starch (ST), sucrose, glucose, and fructose comprising the main components of NSC [10], the growth and survival of plants in natural habitats mostly depend on car-carbon and energy derived from non-structural carbohydrates (NSCs) [10]. A tree’s reaction to environmental elements influences their growth and adaptive tactics; as such, the layout and content of NSC influence these aspects [11]. Understanding how plants adapt to their environment requires studying the impact of environmental conditions on NSC dynamics. This knowledge is crucial for assessing carbon balances at various levels, including organs, individuals, and ecosystems, as well as for understanding ecosystem succession [12,13]. Soluble sugar (SS) in plants mostly serves to enable the transportation and use of carbohydrates, which are essential in controlling cellular osmotic pressure and improving tolerance to hostile environments. ST functions as the main storage material in trees; under certain conditions, it can turn into SS [14]. The functions of various plant organs are specialized, with leaves serving as a crucial site for photosynthesis; stems constituting the primary component of tree biomass; branches acting as vital transport hubs closely associated with the overall C supply of the tree; and roots functioning as one of the principal storage organs for NSC [15]. Research has demonstrated that a low P level stimulates the accumulation of root ST and needles SS in *Pinus massoniana* seedlings [16]; the application of N enhanced the accumulation of SS in both stems and leaves of *Cinnamomun bodinieri* seedlings [17]. The allocation of N and P resulted in a reduction in NSC in the majority of Eucalyptus organs [18]. However, limited prior research has been conducted on the extent of changes in the NSC of sprouts following fertilizer application. Therefore, conducting the study is imperative for effective post-establishment management of *Pinus yunnanensis* Franch. harvesting beds.

*P. yunnanensis* is a major tree species for afforestation in barren mountainous areas of southwestern China [19]. It is vital for ecological restoration in degraded areas, as it stabilizes soils, enhances fertility, and sequesters carbon, all critical for recovering disturbed ecosystems [20]. *P. yunnanensis* exhibits strong ecological adaptability, making it an ideal species for large-scale afforestation projects aimed at restoring ecological balance and enhancing local biodiversity in degraded landscapes [21]. As a pioneer species for forest regeneration, it delivers essential ecosystem services such as erosion control, nutrient cycling, and habitat restoration, playing a pivotal role in recovering degraded forest ecosystems [22,23]. Currently, factors such as human-induced “retrogressive” logging, poor management, natural disasters, and pest outbreaks have exacerbated the degradation of *P. yunnanensis* germplasm resources, negatively affecting local ecological and economic benefits [24]. In forest restoration and the afforestation of degraded areas, clonal propagation techniques offer significant advantages, ensuring stable inheritance of desired traits from parent trees and substantially shortening breeding cycles [25].

However, the maturation impact in the clonal propagation process of *P. yunnanensis* has been shown to be a limiting component for effectively generating propagative materials [26,27]. Previous research have revealed that the cumulative tillering amount of *P. yunnanensis* following coppicing is often limited, maybe due to insufficient NSC reserves or unfavorable nutritional conditions [28,29].

In light of this, the study sought to gather and examine *P. yunnanensis* roots, stems, needles, and sprouts from current year growth over a 270-day period. The goal was to find the distribution pattern of NSC in several organs of *P. yunnanensis* under different fertilizer levels. This work will offer useful data assistance for optimizing fertilizer application following coppicing of *P. yunnanensis*. Moreover, it provides a theoretical basis for obtaining sprouts of *P. yunnanensis* and helps to reveal the growth-adapting processes of *P. yunnanensis* after fertilization application. This work also seeks to close particular knowledge gaps in NSC allocation under fertilization and coppicing.

## 2. Materials and Methods

### 2.1. Plant Material

Within the nursery of Southwest Forestry University (longitude 102°45′41″ E, latitude 25°04′00″ N), the experimental location is located southwest of Kunming City, Yunnan Province, China. About 1945 m above sea level, it has a subtropical semi-humid monsoon climate common of the northern sub-tropical region. With severe lows at −9 °C and highs at up to 32.5 °C, the annual average temperature is 14.5 °C. The area keeps an average relative humidity of 68.2% and receives an annual precipitation between 700 mm and 1100 mm. The seeds used in the experiment came from the Pinus yunnanensis seed orchard located in Midu County, Yunnan Province. The mother trees, from which the seeds were gathered, showed strong growth conditions; the seeds were taken from completely developed cones within the same year. Then, careful screening was carried out to find clearly dried seeds with ideal development.

### 2.2. Experimental Design

To investigate the effects of different nitrogen (N) and phosphorus (P) levels, urea was used as the N fertilizer with a total N content of 46.40%, while superphosphate was used as the P fertilizer with a P_2_O_5_ content of 85%. Two N application levels were selected: 0 g/plant (N_0_) and 0.6 g/plant (N+), to examine the effects of no fertilization and moderate N application on plant growth. The 0.6 g/plant level represents a typical N application rate in conventional forestry management. Three P application levels were chosen: 0.0 g/plant, 2.0 g/plant, and 4.0 g/plant. This choice was made considering P availability in the soil and the needs of the plant, since P plays a major role in the development of *P. yunnanensis*. The 2.0 g/plant level is a common P application rate, while 4.0 g/plant represents a higher P application, aimed at assessing the impact of increased P levels on plant growth. The orthogonal design is an experimental methodology used to systematically investigate the effects of multiple factors (such as different N and P levels) on experimental outcomes and optimize experimental efficiency. This design allows for the effective evaluation of the combined effects of different fertilization levels on *P. yunnanensis* growth without the need for extensive replication. Therefore, the experiment used this approach to establish six treatment groups: D1 (N_0_P_0_), D2 (N+P_0_), D3 (N_0_P), D4 (N+P), D5 (N_0_P+), and D6 (N+P+). Each treatment combination consisted of 35 seedlings replicated three times, totaling 18 plots and 630 seedlings. The seedlings should be decapitated uniformly at the 0 d mark, removing the top 5 cm of height. In accordance with the experimental design, N and P fertilizers were added in March and thoroughly mixed with the substrate at different ratios. The fertilizer was evenly applied twice at the 90 d mark. This experiment on nutrient control was initiated at the beginning of 0 d and finished at the end of 270 d after fertilization.

### 2.3. Material Handling

#### 2.3.1. Sample Treatment

At 0 d, 90 d, 180 d, and 270 d, a number of seedlings, totaling nine, were randomly chosen to be collected, and each treatment was subjected to harvesting. The parts that have been gathered should encompass needles, sprouts, stems, and roots. Subsequently, all collected samples underwent a drying process in an oven at a temperature of 105 °C for 15 min. They were subsequently subjected to additional drying at a temperature of 65 °C until a stable weight was achieved. Following the drying process, the samples were ground using the milling machine (AQ-180E) manufactured in Cixi, China. Finally, they were sieved and securely sealed to facilitate subsequent analysis.

#### 2.3.2. Determination of Soluble Sugar Content

Sugars are one of the essential components of plant structure and serve as primary storage materials for metabolic substrates. The determination was carried out using the anthrone colorimetric method, with assay kits (M1503A) supplied by Suzhou Mengxi Biopharmaceutical Technology Co., Ltd. (Suzhou, China). The experimental procedures followed the instructions provided in the manual. Measurements and calculations were performed according to the method described by Buysse J et al. [30].

#### 2.3.3. Determination of Starch Content

Starch is a nutrient stored in plants, and its content determination is crucial for investigating carbohydrate metabolism within plant tissues. An amount of 80% ethanol was used to separate soluble sugars and starch in the samples. Starch (ST) was then hydrolyzed to glucose using acid hydrolysis, and the glucose content was measured by the anthrone colorimetric method, which allows for the determination of starch content. The assay kit (M1101A) was provided by Suzhou Mengxi Biopharmaceutical Technology Co., Ltd. (Suzhou, China). The experimental procedures followed the instructions provided in the manual. Measurements and calculations were performed according to the method described by Xie et al. [31].(1)$$Non-structural\;carbohydrates\;\left(NSCs\right)=Soluble\;sugar\;\left(SS\right)\;content+Starch\;content\;\left(ST\right)$$(2)$$Phenotypic\;plasticity\;index=\frac{Xmax-Xmin}{Xmax}$$

### 2.4. Statistical Analysis

Data input and collation were performed using Microsoft Excel 2019, while SPSS software was utilized for conducting statistical analysis (version 26.0; IBM Corp., Armonk, NY, USA). Two-way ANOVA was employed to investigate the variations in NSC contents and their components across the entire *P. yunnanensis* plants based on N and P. One-way analysis of variance (ANOVA) was carried out for the purpose of evaluating variances in the contents and components of the NSC among different N, P, and plant organs. Multiple comparisons were carried out using Duncan’s test. The phenotypic plasticity index is used to quantify the degree of variation in different traits of *P. yunnanensis* under different environmental conditions. Graphical representations of the data were generated using Software for analyzing and visualizing data, known as Origin 2022b (Origin Lab, Northampton, MA, USA).

## 3. Results

### 3.1. The Effect of N and P Rationing Pattern on Non-Structural Carbohydrates (NSCs) in Pinus yunnanensis

As illustrated in the diagram provided in Figure 1, at the N_0_ level, the levels of starch (ST) in roots, stems, and sprouts followed the order of D3 > D5 > D1; similarly, the NSC contents in sprouts exhibited a similar trend of D3 > D5 > D1. At the N+ level, SS content in sprouts and NSC content in stems showed significant disparities following the order: D2 > D4 > D6 and D4 > D2 > D6, respectively. Remarkably distinct variations were observed for ST content between stems and sprouts with an order of significance as follows: D4 > D2 > D6, while for sprouts it was found to be D6 > D4 > D2. In essence, without N fertilization present, increasing application of P fertilizer resulted initially in an increase followed by a decrease both for ST content within roots, stems, and sprouts of *P. yunnanensis* as well as total NSC amount within sprouts.

The content of root ST followed this order: D3 > D6 > D4 > D5 > D2 > D1. The content of stem ST followed this order: D4 > D2 > D3 > D5 > D6 > D1; and the content of NSC followed this order: D4 > D2 > D5 > D6 > D3 > D1. In sprouts, the contents of SS were ranked as follows: D2 > D4 > D6 > D3 > D1 > D5; the contents of ST were ranked as follows: D6 > D4 > D3 > D5 > D2 > D1; and the contents of NSC were ranked as follows: D4 > D6 > D2 > D3 > D5 > D1. In conclusion, as fertilizer application increased, the contents of SS, ST, and NSC first climbed and then dropped as fertilizer application increased. This phenomenon implies that more nutrients encourage photosynthesis and carbohydrate buildup in plants, hence improving growth and sprouting capacity at certain fertilizer levels. A nutrient surplus may develop when fertilizer application rises, though, which would cause a restriction in carbohydrate buildup and hence limit the plant’s use of these nutrients. Alternatively, too much nutritional stimulation could change the regulatory systems of several physiological processes, hence lowering SS, ST, and NSC contents. This pattern suggests that fertilization management should be suitably changed depending on plant demands to avoid the negative consequences of over-fertilization, since this may be related to the capacity of the plant to absorb, distribute, and control nutrients for growth.

### 3.2. The Conducting Multivariate Analysis of NSCs in P. yunnanensis

The addition of N had a significant impact on SS and NSC content in *P. yunnanensis* stems, while it also significantly affected the SS, ST, and NSC content in sprouts (Table 1). Furthermore, P addition had a significant effect on the ST content in *P. yunnanensis* stems, as well as on the ST and NSC content in sprouts. Additionally, there was a significant interaction between N addition and P addition that influenced the ST content in *P. yunnanensis* stems (*p* < 0.05).

### 3.3. The Analysis of the Variability of NSCs Among Different Organs of P. yunnanensis

The coefficient of variation for SS content in each organ in response to different levels of N and P addition was as follows: needles exhibited the highest variation, followed by roots, stems, and sprouts. Similarly, the order for ST content and NSC content was needles, which showed the greatest variation, followed by stems, roots, and sprouts. These findings suggest that the allocation of NSC in response to varying degrees of N and P addition primarily occurs in needles (Table 2).

### 3.4. Effects of N and P Addition on the Allocation of NSCs Components to D1

Compared to the D1, fertilization treatments significantly influenced NSC and its components throughout all stages of *P. yunnanensis* growth and development (Figure 2).

At 0 d, roots SS content increased significantly in D2 and D4, while needles’ SS content increased significantly in D5 and D6. Fertilization led to a significant increase in total ST content across all organs of *P. yunnanensis*; however, D4 caused a significant decrease in needle ST content. Sprouts’ NSC content significantly increased after D6. Furthermore, most treatments increased the NSC content and its components in the needles. This indicated that during the early stages of the co-application of N and P, *P. yunnanensis* allocated more carbon to needles for enhanced photosynthesis.

At 90 d, D4 demonstrated an increase solely at the root level, along with sprouts’ SS contents, while experiencing reductions at both stems and needles levels. D5 and D6 displayed an increase across all organ types, including roots, stems, and sprouts, along with a significant decrease specifically within needles’ SS contents. Further, the majority of treatments displayed a noteworthy augmentation in roots’ SS, needles’ ST, and NSC content in both the roots and needles of *P. yunnanensis*.

At 180 d, most treatments led to a significant enhancement in SS and NSC content within each organ of *P. yunnanensis*; meanwhile, D4, D5, and D6 exhibited a remarkable increase in sprouts’ ST.

At 270 d, all other treatments resulted in a considerable rise in NSC content and its constituents throughout all organs. Among them, the majority of treatments showed the greatest increments in stems’ ST and NSC as well as sprouts’ SS and NSC for *P. yunnanensis*.

From the perspective of overall fertilization timing, D4, D5, and D6 showed the best-promoting effects on stem SS and sprout NSC. Therefore, N and P fertilization can stimulate the early allocation of carbon to needles, while also promoting long-term carbon storage in stems and sprouts, thereby supporting the plant’s regenerative ability through sprouting.

### 3.5. Two-Way Analysis of Variance on the Effect of Fertilization on the Soluble Sugar-to-Starch Ratio in P. yunnanensis

The addition of N at 0 d significantly influenced the stem soluble sugar/starch ratio (SS/ST), while the addition of P and the interaction between N and P had no significant effect on the SS/ST of *P. yunnanensis* stems (Table 3). At 90 d, N addition significantly affected the SS/ST of roots, stems, and needles, whereas P addition only had a significant effect on stem SS/ST. Additionally, the N and P interaction significantly influenced stem SS/ST. At 180 d, N addition significantly impacted roots’ and stems’ SS/ST, while P addition had a significant effect on stems’ SS/ST. Moreover, the interaction between N and P significantly influenced the SS/ST of roots, stems, and needles. At 270 d, both N and P additions had a highly significant impact on the sprouts’ soluble SS/ST (*p* < 0.001). Additionally, the interaction between N × P and P showed statistical significance specifically in terms of the SS/ST of *P yunnanensis* stems (*p* < 0.05).

### 3.6. The Effect of N and P Addition on the Soluble Sugar-to-Starch Ratio in P. yunnanensis

At the N_0_ level, the roots’ soluble sugar-to-starch ratio (SS/ST) of *P. yunnanensis* differed significantly, except at 270 d (Figure 3), where the differences were D1 > D3 > D5 (0 d), D5 > D1 > D3 (90 d), and D3 >D5 >D1 (180 d), in order of D1 > D3 > D5. Stems’ SS differed significantly at 180 d and 270 d, in the order of D5 > D3 > D1 and D1 > D5 > D3. In the absence of N fertilizer, the application of P fertilizer progressively increased the SS/ST of roots and stems in *P. yunnanensis* over time, while simultaneously causing a decline in the SS/ST of sprouts. At the N+ level, the SS/ST of stems showed significant differences at 0 d, 180 d, and 270 d, with the order of D2 > D6 > D4 at 0 d and D6 > D4 > D2 at 180 d and 270 d. The SS/ST of needles only exhibited a significant difference at 90 d, with the order of D2 > D6 > D4. The SS/ST of sprouts displayed a significant difference at 270 d, with the order of D4 > D2 > D6. In contrast, the application of P fertilizer does not exert any influence on the SS/ST of roots, whereas it gradually enhances the SS/ST of stems, needles, and sprouts in *P. yunnanensis* over time at the N+ level.

Across all P application levels, the SS/ST in most significantly different organs was higher in N+ than in N_0_.

Furthermore, with the rise in the utilization of fertilizers, the SS/ST of all organs of *P. yunnanensis* during the late stage of fertilization (180 d and 270 d) consistently exceeded 1, indicating a higher proportion of SS/ST. The observation suggests that in post-fertilizer application, *P. yunnanensis* predominantly accumulates NSC in the form of SS. This also implied that after fertilization, the growth of all organs of *P. yunnanensis* seedlings became more vigorous.

### 3.7. Correlation Analysis of Sprout Growth and NSCs in P. yunnanensis Seedlings

Spearman correlation analysis reflected the mechanism of the effect of NSC on the growth quality of *P. yunnanensis* seedlings (Figure 4). The results showed that the cumulative tillering amount was significantly positively correlated with sprout NSC (r = 0.677, *p* < 0.001) and needle SS/ST (r = 0.572, *p* < 0.001), while it was significantly negatively correlated with sprout SS/ST (r = −0.358, *p* < 0.001) and sprout ST (r = −0.285, *p* < 0.05). Initial sprouts were significantly positively correlated with sprout NSC (r = 0.308, *p* < 0.05) but significantly negatively correlated with sprout ST (r = −0.476, *p* < 0.001) and sprout NSC (r = −0.342, *p* < 0.05). Potential sprouts were significantly positively correlated with root SS (r = 0.416, *p* < 0.001), needle ST (r = 0.389, *p* < 0.001), and needle NSC (r = 0.326, *p* < 0.05), but significantly negatively correlated with needle SS/ST (r = −0.416, *p* < 0.001). Effective sprouts were significantly positively correlated with sprout NSC (r = 0.535, *p* < 0.001) and significantly negatively correlated with stem SS (r = −0.523, *p* < 0.001) and sprout SS/ST (r = −0.365, *p* < 0.001).

Mantel test correlation analysis indicated that the cumulative tillering amount, potential sprouts, effective sprouts, stem SS, needle SS, root ST, stem ST, sprout ST, root NSC, stem NSC, needle NSC, sprout NSC, stem SS/ST, needle SS/ST, and sprout SS/ST were significantly correlated with fertilization. Among these, sprout ST had the strongest positive correlation with N fertilization alone, while sprout NSC showed a strong positive correlation with P fertilization alone and N and P co-fertilization. Sprout NSC also had a strong positive correlation with N and P co-fertilization. Sprout SS/ST had the strongest negative correlation with P fertilization alone.

In summary, Spearman correlation and Mantel test analysis emphasized the key role of NSC in shaping the growth and sprouting dynamics of *P. yunnanensis*. Fertilization, particularly with N and P, significantly influenced the accumulation of NSC in different organs, thereby supporting growth and regeneration.

### 3.8. Phenotypic Plasticity Analysis of Sprout Growth and NSCs in P. yunnanensis

From Figure 5, it was observed that among all plasticity indicators, the plasticity index of root NSC was the lowest, while the plasticity index of cumulative tillering amount was the highest. In terms of the growth traits, the plasticity index of cumulative tillering amount (0.7692) was greater than 0.5, indicating the highest plasticity. For sprout length, effective sprouts had a plasticity index (0.7015) greater than 0.5, showing the highest plasticity, while the plasticity of initial and potential sprouts’ parameters were less than 0.5. In terms of NSC, the plasticity indices of all parameters were less than 0.5, with three parameters showing relatively higher plasticity: stem SS/ST (0.4455), stem ST (0.4100), and sprout SS/ST (0.3876). The root NSC content showed the lowest plasticity (0.1655). These results indicated that after coppicing, the response of *P. yunnanensis* seedlings to fertilization was primarily through changes in stem SS/ST, stem ST, and sprout SS/ST to enhance sprout growths.

## 4. Discussion

The changes in the levels of carbohydrates within plant bodies, as well as their interconversion, are important physiological responses for plants to adapt to external environments and maintain their primary functions such as growth, respiration, reproduction, and defense [32]. It is also the balanced response of plants between photosynthesis and the process of growth respiration [33]. Plant productivity is significantly influenced by the allocation of carbohydrates, which can exhibit variations depending on different life forms [34]. Plants contain two types of carbohydrates, namely structural and non-structural. Structural carbohydrates refer to polysaccharides found in the cell wall, which play a crucial role in providing support to the plant’s structure [35], whereas the reserves in plant tissues, namely SS and ST, primarily constitute the non-structural carbohydrates (NSCs) [36].

While the interaction between nitrogen (N) and phosphorus (P) addition had no significant effect on the NSC content in the organs of *Pinus yunnanensis*, our results revealed that N addition considerably altered the SS, ST, and NSC content in the sprouts of *P. yunnanensis* (*p* < 0.05; Table 1). P can promote plant growth, but it may lead to excessive energy consumption [37]. At the N_0_ level, the ST content in the roots, stems, and sprouts of *P. yunnanensis* increased with the increasing P application, while the NSC content in the sprouts first increased and then decreased as P application increased. This suggested that with increasing P application, the growth rate of sprouts accelerated, consuming energy that would otherwise be used for NSC storage, thereby reducing NSC accumulation. This suggested that while P fertilization had clear advantages in enhancing plant growth, excessive use may accelerate certain physiological processes, leading to excessive energy consumption and affecting the overall health of the plant. P fertilizers should thus be used carefully to balance their advantages with possible drawbacks. When nutritional levels are sufficient, ST, as a reserve carbohydrate, can be quickly used to support development and metabolic needs, therefore reducing the starch content in some plant organs [38,39]. Our work revealed that increasing P application following N addition reduced the ST content in *P. yunnanensis* stems and sprouts. This implied that suitable fertilization could support the fast development of *P. yunnanensis* stems and sprouts, hence boosting the plant’s ability for regeneration and recovery. In the carbohydrate metabolism of a plant, starch (ST) and soluble sugars (SS) are dynamically balanced overall [40]. Plants that have enough fertilizer give more carbs top priority when turning them into SS to use them quickly, instead of storing them as ST [41]. Given that plants must rapidly use soluble sugars to maintain metabolism and cell development, this conversion process may be related to their fast adaptation to the external environment, particularly in nutrient abundance [42].

At the P_0_ level, with the increase in N addition, the NSC content of *P. yunnanensis* sprouts showed an increasing trend, while the SS and ST content of other organs remained relatively stable, as well as P and P+ treatment (Figure 1), which means that N treatments can have a large effect on aboveground wood NSC concentrations [43]. The same phenomenon was also noted in the study conducted by Liu et al. [44]. The possible reason for this phenomenon could be attributed to the fact that elevated N inputs into the soil resulted in an increase in plant NSC content and carbon fixation [45]. A global meta-analysis conducted on a global scale indicated that the addition of N could potentially augment net primary productivity through its impact on attributes associated with the process of photosynthesis, encompassing the overall extent of leaf surface area, rate of photosynthesis per leaf area, and rate of transpiration [46]. Meanwhile, under conditions of moderate drought, it is possible for the NSC levels to rise as a result of the decoupling between photosynthesis and growth [47,48]. The observed phenomenon may be attributed to the alteration of nutrient allocation pattern in *P. yunnanensis* caused by leveling stubble [49]. The simultaneous application of fertilizer supplied ample nutrients to the seedlings, thereby facilitating the germination of *P. yunnanensis* seedlings [50,51]. This dynamic highlighted the broader ecological context in which N input and management practices such as coppicing lead to NSC accumulation [51,52]. Compared to studies on pines or other conifers, we observed a similar pattern, where N fertilization typically increases carbohydrate reserves in the aboveground tissues. Nevertheless, species-specific reactions and environmental factors could affect the degree of these effects [53,54,55]. Understanding plant physiological processes depends on finding the balance between carbon fixation and the distribution of carbohydrates to various organs, especially in relation to changing nutrition supplies [56]. This underlines the need to take nutrient availability into account while deciphering NSC trends and their ecological relevance in plants.

Our study found that the content of SS and NSC in sprouts of *P. yunnanensis* sprouts increased significantly with increasing N addition under treatment involving levels of P_0_ and P addition (Figure 2). In contrast, Liu et al. found that N supply had a greater impact on enhancing plant growth rather than storing NSC, resulting in a decline in the concentration of NSC in *Camellia japonica* [57]. The decline in NSC concentration was attributed to the prioritized stimulation of plant growth rather than the storage of NSC induced by N supply. Thus, the promotion of plant growth by N supply was prioritized over NSC storage, resulting in a reduction in the concentration of NSC in Camellia japonica [58,59]. This contradicts the findings of our research. The observed occurrence could be ascribed to the impact of N and P fertilization on initial photosynthesis and the distribution of carbohydrates between source and sink in plant tissues [60]. Introducing P into the soil could potentially enhance rates of nitrification and/or N_2_ fixation, leading to an overall increase in N availability for plants in soils with a deficiency of P. Conversely, the inclusion of N may reduce the availability of soil P, as it induces acidification [61,62]. Conversely, regarding the P_0_ and P treatments, the P+ treatment exhibited a reduction in stem NSC content with an increasing N application rate, indicating a significant increase in NSC due to N application. This finding aligns with the results reported by Zhang et al. [63], and is similar to the results of Zhang et al. [10]. The prolonged application of N fertilizers is likely to be the primary factor contributing to soil acidification [64]. Therefore, the careful management of N and P fertilization is crucial to prevent the negative impacts of excessive fertilization on soil nutrient balance and ecosystem health.

The distribution of NSC among various plant organs depends on their individual capacities (or competing needs) for NSC acquisition and utilization. Generally, plants adhere to the “priority-based distribution principle” by reorganizing the allocation of carbon resources to vital organs that are of utmost importance, and those given priority are those undergoing growth for the provision of NSCs [65,66]. Previous studies have shown that seedlings need a lot of nutrients for sprouting growth after topping, and fertilizer application can effectively provide the nutrients needed for seedling and sprouting growth after topping, and improve the quality of sprouting and sprouting ability [67,68]. In the research conducted by us, we have discovered that the application of fertilizers increased the NSC and its components in the sprouts of *P. yunnanensis* over time, which was attributed to the fact that fertilizer application promoted the development of photosynthetic organs and accelerated the production and transport of photosynthetic products [69]. Meanwhile, the decrease in needle ST content in D4 indicates that the application of the appropriate amount of fertilizer can lead to the timely conversion of ST to SS for plant growth, which echoes the most significant increase in the content of SS in D4 of the present study. We also found that the content of SS, ST, and NSC increased in all organs of spruce pine seedlings after a certain time of growth after fertilization, and this result was different from that of spruce [70]. The growth period of the seedlings in this experiment may have been slowed down by the supplementary fertilization, resulting in a continuous increase in SS, ST, and NSC content. Additionally, as the temperature decreased during the season, the seedlings accumulated a significant amount of SS to enhance their cold resistance and maintain normal growth [71].

The arrangement of NSC constituents is an outcome of the collaborative effect of biological functions, showcasing plants’ ability to adjust to various environmental factors [72,73] that have a strong correlation with the strategies plants employ for their survival [74]. While ST offers advantages in terms of plants’ ability to store energy over extended periods [75], SS plays a crucial role not only in cellular osmotic adjustment but also functions as a signaling molecule facilitating the adaptation to alterations in the surroundings [76]. We found that the distribution of SS and ST in specific organs of the plant reflected inter-organ translocation at different levels of fertilization. At the beginning of the fertilizer application, the ratio of soluble sugar-to-starch ratio (SS/ST) was less than 1 (Figure 3), which may be due to the rapid growth of seedlings after fertilizer application, thus leading to the consumption of large amounts of SS [77]. The research carried out by Kong et al. demonstrated that the majority of tissues exhibited an SS/ST ratio exceeding 1, providing evidence for SS being the predominant form of NSC [18]. Given that ST functions as a carbon reserve, whereas SS exhibits multiple functionalities [78], the high SS/ST ratio in the different organs is due to intense metabolic activities [79]. This provides strong support for our analysis of SS and ST.

In our study, there is a significant correlation in the growth of sprouts of *P. yunnanensis* seedlings under fertilization conditions (Figure 4). This indicates that the trade-off between growth and NSC storage in coppiced *P. yunnanensis* seedlings differs after fertilization. Studies have shown that NSC content in stems significantly decreases after coppicing [80,81], and our study found similar results: the higher the NSC content in sprouts of *P. yunnanensis* seedlings after coppicing, the greater the number of effective sprouts; in contrast, stem SS and the SS/ST in sprouts negatively regulate the number of effective sprouts, indicating that the carbon needed for sprout growth in *P. yunnanensis* seedlings is supplied by the stem. Meanwhile, the conversion of SS to ST in the sprouts is more conducive to sprout growth.

Throughout their long evolutionary processes, plants continuously adapt to and are selected by their surrounding environments, eventually forming various internal physiological and external morphological adaptive strategies (Figure 5). These ecological adaptive traits are referred to as plant traits [82]. Plants rely on their own phenotypic plasticity to adjust the acquisition and consumption of various resources in nature to maintain their normal growth and metabolism, thereby responding to different environmental conditions [83] The adaptive regulation of *P. yunnanensis* seedlings under fertilization is a complex physiological process. Plasticity analysis revealed that the plasticity index of various root traits in *P. yunnanensis* seedlings is relatively low, while the plasticity index of the cumulative tillering amount, effective sprouts, sprout SS/ST, and stem SS/ST are relatively high. The cumulative tillering amount and effective sprouts reflect the ability of *P. yunnanensis* seedlings to produce high-quality sprouts after fertilization. The SS/ST in the sprouts and stems reflects the ability to maintain the dynamic balance of NSC in sprouts and stems. Under fertilization, *P. yunnanensis* seedlings mainly improve the quality of the sprouts through carbon transport from the roots and stems. Overall, fertilization improved plant quality and yield, which is consistent with the findings of many researchers [84,85,86,87].

## 5. Conclusions

The application of nitrogen (N) and phosphorus (P) fertilizers significantly affects non-structural carbohydrates (NSCs) and their components’ contents in the organs of *Pinus yunnanensis* seedlings. As fertilization increased, NSCs and their components’ contents in *P. yunnanensis* seedlings first increased and then decreased, all significantly higher than in D1. Additionally, NSCs and their components’ contents in the roots was significantly higher than in other organs. In D4, the root SS and ST contents were the highest. The ST and NSC contents were highest in the stems and sprouts. In D6, SS content in the stems and sprouts, and the NSC contents in the needles, were the highest.

From a temporal perspective, fertilization significantly increases NSCs and their components’ contents in *P. yunnanensis* seedlings. These findings suggest that the roots play a crucial role in regulating the distribution of NSCs in *P. yunnanensis*.

In summary, an appropriate fertilization strategy can promote a balance between nutrient acquisition and storage in *P. yunnanensis* after coppicing, enhance sprouting ability, and further promote sprout growth. For future reforestation and afforestation projects, it is recommended to continue using N-P combined fertilization methods, while carefully monitoring the impact on soil health, especially soil pH, as long-term fertilization may lead to soil acidification. This work underlines the need of fertilization in fostering good development and sprouting after coppicing in afforestation methods and offers important new perspectives on the dynamics of NSCs in *P. yunnanensis*. These results can, in the meantime, provide direction for fertilization techniques in major initiatives focusing on forest restoration, especially in nutrient-deficient regions.

## Figures and Tables

**Figure 1 plants-14-00462-f001:**
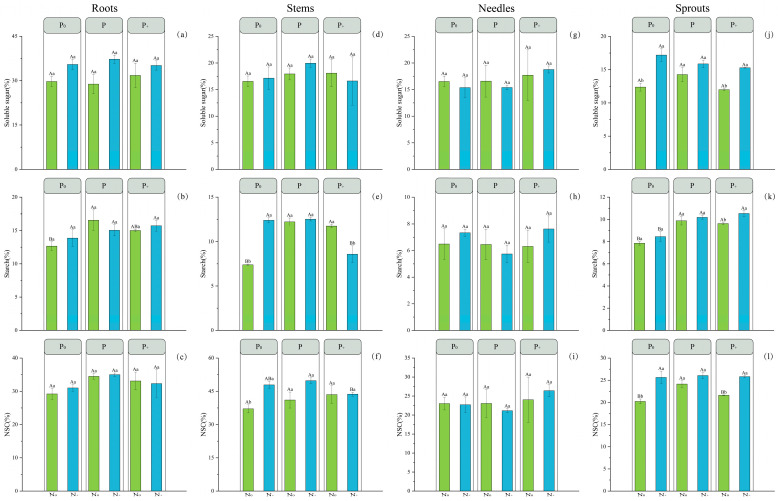
Effects of variations in the allocation of N and P on non-structural carbohydrates and their components’ contents were investigated in *P. yunnanensis* at 270 d. The levels of soluble sugar (**a**,**d**,**g**,**j**), starch (**b**,**e**,**h**,**k**), and the overall NSC pools (**c**,**f**,**i**,**l**) were measured in different organs of *P. yunnanensis* and the treatments were named as follows: N_0_P_0_ (D1), N+P_0_ (D2), N_0_P (D3), N+P (D4), N_0_P+ (D5), and N+P+ (D6). Error bars represent ±1 S.E. with a sample size of 3. Please note that the data presented in the figure are expressed as mean values ± standard error (N = 9). Distinct capitalized characters indicate significant differences observed among P allocation patterns under identical N allocation patterns at a significance level (*p* < 0.05); while distinct lowercase letters denote significant differences between various P allocation patterns within the same N allocation pattern using a level of significance (*p* < 0.05).

**Figure 2 plants-14-00462-f002:**
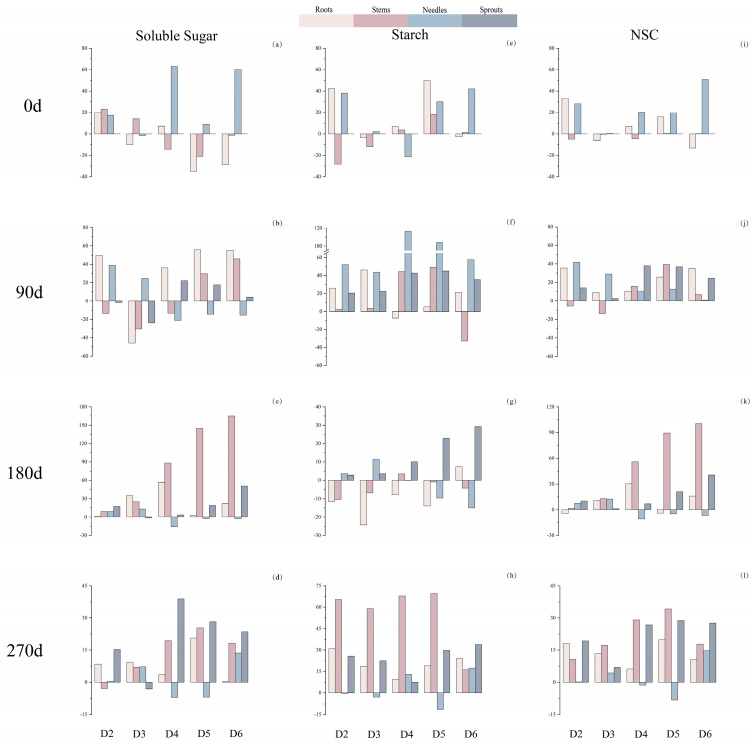
Effects of N and P rationing (%) on non-structural carbohydrates and their components contents under different treatments (D2, D3, D4, D5, D6) compared with D1 in four determination periods (0 d, 90 d, 180 d, 270 d). Soluble sugar (**a**–**d**), starch (**e**–**h**), and NSC (**i**–**l**) in *P. yunnanensis*.

**Figure 3 plants-14-00462-f003:**
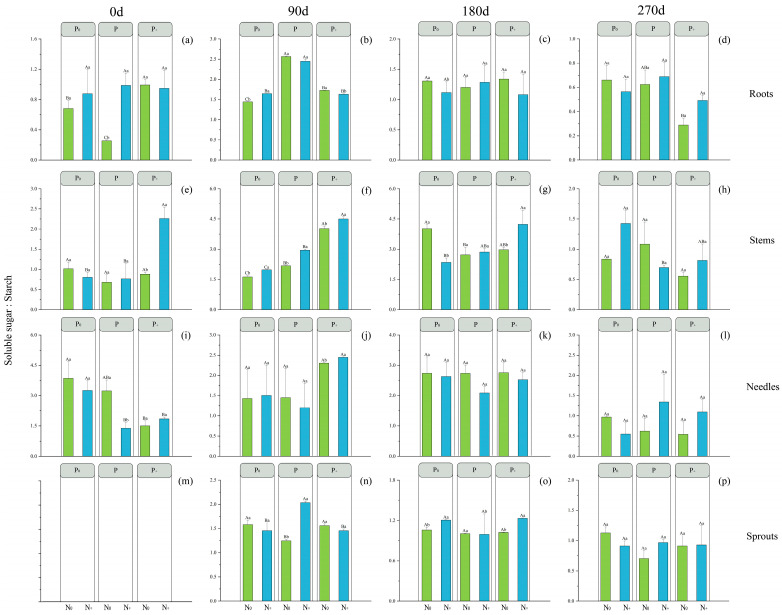
The effect of the N and P rationing pattern on the soluble sugar/starch ratio (SS/ST) (the same below) in *P. yunnanensis*. Roots (**a**–**d**), stems (**e**–**h**), needles (**i**–**l**), and sprouts (**m**–**p**) are shown, and the treatments were named as follows: N_0_P_0_ (D1), N+P_0_ (D2), N_0_P (D3), N+P (D4), N_0_P+ (D5), and N+P+ (D6). The data depicted in the figure are represented as the average ± standard error. (N = 9). The significant differences in P rationing pattern amount under the same N rationing pattern are indicated by distinct uppercase letters (*p* < 0.05); while significant differences between different P rationing patterns at the same N rationing pattern are denoted by distinct lowercase letters (*p* < 0.05).

**Figure 4 plants-14-00462-f004:**
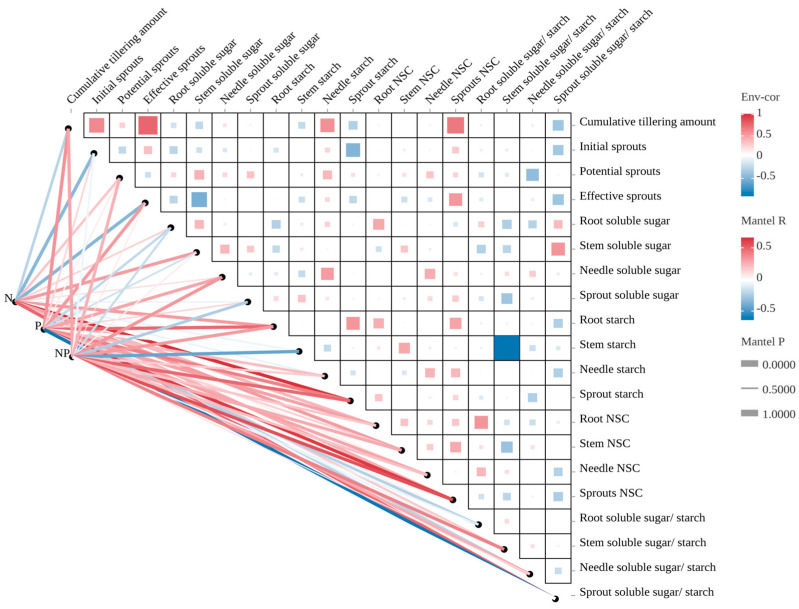
A network heatmap of correlations between fertilization between sprout growth and non-structural carbohydrates and their components’ contents traits in *P. yunnanensis.* Note: Initial sprouts: 0 cm < sprouts length ≤ 2 cm; potential sprouts: 2 cm < sprouts length ≤ 5 cm; effective sprouts: 5 cm < sprouts length.

**Figure 5 plants-14-00462-f005:**
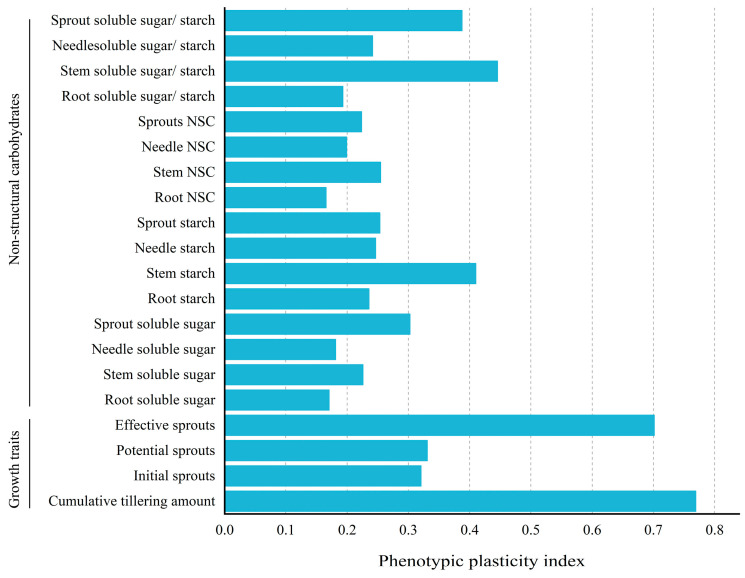
Phenotypic plasticity index of sprout growth, non-structural carbohydrates and their components’ contents, and soluble sugar-to-starch traits across different organs in *P. yunnanensis.*

**Table 1 plants-14-00462-t001:** The importance of conducting multivariate analysis of variance to assess the impact of N rationing, P rationing, and organ variations on non-structural carbohydrates and their components contents. (*F*-value).

Factor	Organ	Soluble Sugar	Starch	NSC
N	Roots	0.293	0.021	0.286
	Stems	13.458 **	3.794	9.904 **
	Needles	0.524	0.006	0.227
	Sprouts	34.097 ***	6.950 *	39.677 ***
P	Roots	0.463	2.849	1.418
	Stems	0.001	16.663 ***	0.710
	Needles	0.162	0.396	0.005
	Sprouts	2.466	31.180 ***	4.300 *
N × P	Roots	0.454	0.953	0.112
	Stems	1.002	41.466 ***	2.597
	Needles	0.084	1.200	0.318
	Sprouts	2.794	0.553	2.826

Note: *: *p* < 0.05, **: *p* < 0.01, ***: *p* < 0.001. N: N addition; P: P addition.

**Table 2 plants-14-00462-t002:** Distribution characteristics of non-structural carbohydrates and their components’ contents in various organs of *P. yunnanensis.*

Index	Organs	Min. (%)	Max. (%)	Mean. (%)	Standard Deviation	Coefficient of Variation (%)
Soluble sugar	Roots	10.238	25.397	17.721 b	3.681	20.772
	Stems	24.497	39.720	33.036 a	4.878	14.765
	Needles	10.752	26.81	16.733 b	3.806	22.745
	Sprouts	11.394	18.974	14.499 b	2.148	14.814
Starch	Roots	11.519	18.205	14.812 a	1.921	12.969
	Stems	6.786	12.959	10.821 b	2.205	20.377
	Needles	4.199	9.396	6.662 c	1.549	23.251
	Sprouts	7.566	10.812	9.425 b	1.062	11.268
NSC	Roots	25.662	39.893	32.533 b	3.797	11.671
	Stems	34.783	52.162	43.857 a	5.633	12.844
	Needles	15.659	35.547	23.395 c	4.842	20.697
	Sprouts	19.546	28.150	23.924 c	2.546	10.642

Note: Dissimilar letters denote statistically significant distinctions at a significance level of 0.05 across various organs. *p* < 0.05.

**Table 3 plants-14-00462-t003:** A two-way analysis of variance (ANOVA) was conducted to examine the impact of different N and P rationing patterns on the soluble sugar-to-starch ratio of *P. yunnanensis* during four specific time periods (*F*-value).

Time	Factor	Soluble Sugar-to-Starch			
		Roots	Stems	Needles	Sprouts
0	N	2.456	6.312 *	0.840	-
	P	3.150	0.670	1.346	-
	N × P	1.167	1.744	0.299	-
90	N	6.291 *	6.577 *	23.410 ***	0.024
	P	1.176	4.837 *	0.138	0.987
	N × P	1.308	9.454 **	0.686	0.686
180	N	11.429 **	943.868 ***	1.277	0.004
	P	234.967 ***	100.123 ***	0.769	0.352
	N × P	1318.119 ***	22.982 ***	0.664	1.430
270	N	0.024	1.301	0.684	14.933 **
	P	0.275	3.323	0.330	16.420 ***
	N × P	0.316	7.290 **	0.571	2.531

Note: *: *p* < 0.05, **: *p* < 0.01, ***: *p* < 0.001. N: N addition; P: P addition.

## Data Availability

Data are available upon reasonable request.

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
