# Peer review of "Nitrogen and Phosphorus Co-Fertilization Affects Pinus yunnanensis Seedling Distribution of Non-Structural Carbohydrates in Different Organs After Coppicing"

_plants, 2025, doi:10.3390/plants14030462_

Round 1
Reviewer 1 Report
Comments and Suggestions for Authors
The manuscript investigates the effects of nitrogen (N) and phosphorus (P) co-fertilization on the distribution of non-structural carbohydrates (NSCs) in Pinus yunnanensis seedlings following coppicing. This study is well-timed given the importance of understanding how fertilization can enhance growth and adaptive mechanisms of forest species, especially in degraded environments. The use of an orthogonal experimental design and detailed NSC measurements is commendable. The study addresses a significant ecological and silvicultural challenge—how to optimize fertilization strategies for Pinus yunnanensis seedlings, a key species in reforestation efforts in Southwest China. The authors employ a systematic experimental design with various N and P levels and an orthogonal setup, allowing for a robust analysis of factorial interactions. The manuscript provides a comprehensive dataset, detailing changes in NSC components (soluble sugars and starch) across different organs and time points. References to previous studies on nutrient addition and carbohydrate storage in plants provide a solid background.
However, there are notable areas for improvement in the manuscript's structure, methodology, interpretation, and clarity.
(1) The abstract lacks clarity and precision. Although the major findings are stated, the flow of ideas is difficult to follow. It also includes redundant phrases and excessive technical jargon, making it less accessible. So please revise the abstract to emphasize the context, key findings, and their significance concisely. Avoid unnecessary repetition.
(2) The introduction provides a good overview of N and P fertilization's ecological importance but lacks focus on Pinus yunnanensis. Additionally, it does not clearly justify why NSC dynamics post-coppicing are particularly relevant for this species. So please expand on why Pinus yunnanensis is a crucial species for afforestation in degraded areas. And highlight specific knowledge gaps in NSC allocation following coppicing and fertilization that this study aims to fill.
(3) The description of the experimental design is overly technical in some places, which may confuse readers unfamiliar with the methodology. Additionally, details regarding the replication strategy and statistical power are limited. So, please provide a simplified explanation of the orthogonal design and clarify the reasoning behind the specific levels of N and P used. Elaborate on the replication strategy and statistical assumptions.
(4) The procedures for NSC quantification are briefly mentioned but not sufficiently detailed. There is also no mention of potential biases or sources of error in these measurements. So, please include a brief justification for the choice of NSC analysis methods and address potential limitations (e.g., how drying and grinding may affect the results).
(5) The results section is overly dense and repetitive, making it challenging to identify the key findings. Figures are difficult to interpret due to inconsistent labeling and an overwhelming amount of data presented at once. So, please use subheadings to structure the results more clearly (e.g., "Effect of N and P on NSC Allocation by Organ"). And simplify figures by focusing on the most critical comparisons and trends.
(6) The interpretation of the statistical results is inconsistent. For example, certain significant trends are described, but their biological relevance is not always explained. So please clearly connect statistical significance to biological significance. For instance, explain how increases in root NSC content translate to improved seedling growth.
(7) The discussion reiterates results without providing deeper insights into the ecological or practical implications of the findings. The potential mechanisms behind the observed trends in NSC allocation are not well-explored. So please discuss the physiological mechanisms driving the observed NSC patterns in different organs. And compare findings with similar studies on Pinus species or other conifers to provide broader ecological context.
(8) While the study aims to inform fertilization strategies, practical recommendations for forest managers are absent. So, please provide concrete recommendations for fertilization regimes in reforestation projects based on the study's findings.
(9) Many figures (e.g., heatmaps and bar graphs) are overcrowded with data, making them difficult to interpret. Some tables are redundant, with data that could be integrated into the text. So please simplify figures and ensure axes and legends are clear. And consolidate tables and figures where possible to avoid redundancy.
(10) The conclusion is overly broad and does not sufficiently emphasize the study's novel contributions. So, please provide a focused summary of the study's most significant findings and their implications for reforestation practices. Highlight how the research contributes to the broader understanding of NSC dynamics in forestry.
(11) The manuscript contains numerous grammatical errors and awkward phrasing, which reduce its readability. So please conduct a thorough proofreading or seek professional editing to improve grammar and clarity. For example, simplify technical terms where possible.
Reviewer 2 Report
Comments and Suggestions for Authors
The work contains issues that are important from the point of view of plant growth and development.
However, I believe that there are major shortcomings from the methodological point of view.
The study used 1 dose of N and 2 doses of P, because the dose of No and Po - this is the control - concerns the growth of the plant in a medium that was enriched with a nutrient - there is no detailed information on this subject or the composition of this medium.
The physicochemical parameters of the substrate should be presented.
From the analytical side – there are no details on how SS and ST were determined.
Date of collection should be unified (months in the methods section, days in the results section and on the graphs).
Dose description (combinations) should be unified – sometimes there is CK, D2, D3 … sometimes there is (N0P0 treatment), (N+P0), (N0P) – this makes the work difficult to read.
The graphs are difficult to read – there is no consistency in the graphical aspect.
Statistics in the work is very extensive and presented in a complicated way - this should be simplified.
The aim of the work does not fully correspond to the content of the work, the content of the work is more complex than the assumed goals.
I do not deny the enormous amount of work that the authors put into the research – however, the work needs to be organized and refined. Made more readable. Statistics overshadowed the aim of the work.
Comments on the Quality of English LanguageThe English language should be verified by specialists.
Reviewer 3 Report
Comments and Suggestions for Authors
Dear Authors,
Your manuscript entitled “Nitrogen and Phosphorus Co-fertilization Affects Pinus yunnanensis Seedlings Distribution of Non-structural Carbohy- 3 drates in Different Organs After Coppicing ” Authors: He Sun, Yu Wang , Lin Chen , Nianhui Cai , Yulan Xu, is nice written. Paper makes a significant contribution to science. I suggest that the paper be accepted for publication in Sugar Tech after minor corrections give to the paper.
Abstract
To investigate the effect of nutrient additions on the non-structural carbohydrates (NSCs) of Pinus yunnanensis Franch., the different fertilizer proportion was applied to the current year's growth of P. yunnanensis after coppicing in March 2022. The experiment employed two levels of nitrogen (N) addition and three levels of phosphorus (P) addition for orthogonal experimental design to comparatively investigate the effects of different N and P additions on the content of NSCs and their components in various organs. The findings demonstrate that fertilization benefits the promotion of NSC and their component accumulation in diverse organs. Applying P fertilizer can significantly enhance the stem and sprout starch (ST) content and the NSCs content in sprouts. The application of N fertilizer significantly increased soluble sugar (SS) and the NSCs content in the stems and the content of NSC and their components in sprouts. Simultaneously applying N and P can significantly increase the stem SS content. Compared to CK, fertilization had varying effects on the levels of NSC and its components of P. yunnanensis at different time points. At 0 d, NSC and their components significantly increase in the needles after fertilization. At 90 d, the roots and stems exhibited an augmentation in both NSC levels and their constituent components. At 180 days, the ST content in the stems experienced a notable increase. At 270 d, there was a significant increase in NSC and its components across all organs. In terms of integrated fertilization timing, the N (0.6 g/ 28 plant) and P (2.0 g/ plant) treatment (D4) exhibited the highest content of roots SS, roots, stems, and sprouts ST and NSC. After coppicing, fertilization affects NSCs of P. yunnanensis seedings. Notably, the integrated treatment of D4 demonstrated a pronounced effect on promoting NSC accumulation in P. yunnanensis.
Keywords: Pinus yunnanensis; coppicing; non-structural carbohydrate; fertilization; nitrogen; phosphorus
Comment:
1. In abstract in line 16 In the abstract, add the values of the fertilization levels used in the experiment.
2. I ask the authors to add a description and significance of Pinus yunnanensis in the introduction.
3. In Material and methods Indicate by which methods you performed analyzes for soluble sugar (SS), NSC contents and starch (ST).
4. And add full name for NSC contents in
5. In table 1 in line 193, 5 page add full name for NSC contents.
6. In table 2 in line 210, 6 page add full name for NSC contents.
7. In figure 2 in line 254, 7 page add full name for NSC contents.
8. In line 477 add next:
Many researchers have reached similar results [68-71] and state that by applying fertilization, plants mainly improve the quality and yield.
9. Please expand the conclusion, add research results.
10. In references il line 534 add new citate...
68. Stevanović A., Popović V., Filipović A., Bošković J., Pešić V., Marinković J., Stojićević A. 2024. Pytopharmacological profile, nutritional value and amaranthine content of Amaranthus and their significant in medicine. Notulae Botanicae Horti Agrobotanici Cluj-Napoca, 52, 4, 14070. https://doi.org/10.15835/nbha524140070
71. Popovic V., Vasileva V., Ljubičić N., Rakaščan N., Ikanović J. Environment, soil and digestate interaction of maize silage and biogas production. Agronomy. 2024, 14 (11), 2612; https://doi.org/10.3390/agronomy14112612
I ask the authors to adopt the suggestions in order to improve the paper. Thank you.
